# Towards a Holistic Model Explaining Hearing Protection Device Use among Workers

**DOI:** 10.3390/ijerph19095578

**Published:** 2022-05-04

**Authors:** Olivier Doutres, Jonathan Terroir, Caroline Jolly, Chantal Gauvin, Laurence Martin, Alessia Negrini

**Affiliations:** 1Department of Mechanical Engineering, École de Technologie Supérieure (ÉTS), 1100 Rue Notre-Dame Ouest, Montréal, QC H3C 1K3, Canada; 2Institut National de Recherche et de Sécurité (INRS), 1 Rue du Morvan, 54500 Vandoeuvre-lès-Nancy, France; jonathan.terroir@inrs.fr; 3Institut de Recherche Robert-Sauvé en Santé et en Sécurité du Travail (IRSST), 505 Boul. De Maisonneuve Ouest, Montréal, QC H3A 3C2, Canada; caroline.jolly@irsst.qc.ca (C.J.); chantal.gauvin@irsst.qc.ca (C.G.); alessia.negrini@irsst.qc.ca (A.N.); 4Faculty of Medicine, Université de Montréal, 7077 av. du Parc, Montréal, QC H3N 1X7, Canada; laurence.martin.1@gmail.com

**Keywords:** hearing protection device, comfort model, noise-induced hearing loss prevention

## Abstract

Offering hearing protection devices (HPDs) to workers exposed to hazardous noise is a noise control strategy often used to prevent noise-induced hearing loss (NIHL). However, HPDs are used incorrectly and inconsistently, which explains their limited efficiency. Numerous models based on social cognition theories identify the significant factors associated with inconsistent HPD use and aim to improve HPD training programs and to increase HPD use. However, these models do not detail (dis)comfort aspects originating from complex interactions between characteristics of the triad “environment/person/HPD” while these aspects are known to largely influence HPD (mis)use. This paper proposes a holistic model explaining HPD (mis)use, based on the integration of a comfort model adapted to HPDs into an existing behavioral model already developed for HPDs. The model also takes into account the temporal dimension, which makes it possible to capture the scope of change in HPD-related health behaviors. This holistic description of HPD use could be used as a tool for stakeholders involved in HPD use to effectively prevent NIHL among workers.

## 1. Introduction

In order to prevent permanent occupational noise induced hearing loss (NIHL), two engineering noise control strategies are commonly recommended as a first line of defense: modifications or replacement of the noise sources and noise reduction along the transmission path using, for example, noise barriers or sound absorbing materials. However, to date, these noise control strategies fail to sufficiently reduce noise exposure to a safe level for many workers. The use of hearing protection devices (HPDs) is then recommended to protect those workers. Typical HPDs used in the industrial sector are earplugs which are inserted in the earcanals and earmuffs which are worn over the ears [1]. However, the effectiveness of this last strategy is questionable [2,3,4,5] since HPDs are sometimes not used at all or used inconsistently and/or incorrectly. Understanding these unhealthy behaviors and their origins is therefore of utmost importance in order to propose solutions to improve the efficiency of this means of protection against NIHL.

### 1.1. Understanding HPD Consistency of Use through Behavioral Models

Numerous research studies investigate the hearing protection behavior of workers in different sectors (i.e., construction, factory, agriculture, entertainment or service, mining, firefighting and military), aiming to identify the main factors (predictors) related to workers’ inconsistent use of HPDs and contribute to develop effective interventions to promote HPD use. They rely on behavior change theories or health behavior models such as the Health Promotion Model (HPM) [6,7,8,9,10,11,12,13,14,15,16,17,18,19,20,21,22,23,24,25,26,27,28,29,30,31,32,33,34], the Health Belief Model (HBM) [13,18,22,28,29,32,35,36,37,38,39,40], the Theory of Reasoned Action (TRA) or Theory of Planned Behavior (TPB) [13,22,35,39,41], Transtheoretical/Stages of change models [6,13,14,22,24,35,39,42,43,44], the Ecological Model for Health Prevention [45,46,47,48,49], the Protection Motivation Theory (PMT) [36,50], the Extended Parallel Process Model (EPPM) [51,52,53] or the Rational-Emotive Behavior Therapy (REBT) [40,50]. These models are differentiated by the variables studied to explain why people fail or succeed when it comes to engage in a preventative health behavior. Among the aforementioned models, those based on social cognition theory and models (e.g., HPM, HBM, TRA, TPB) are the most widely used and proven to predict the consistent use of HPDs. Overall, these studies reveal that this HPD-related health behavior has multiple and complex origins. Nevertheless, two cognitive-perceptual factors are more often pointed out as important predictors influencing HPD use: “perceived barriers” [6,7,8,9,11,12,13,15,19,20,21,22,25,26,27,29,33,40,46,47,50,53,54,55,56,57,58,59] which describes the impediments to the use of HPDs, and “perceived benefits” [6,7,8,9,12,15,16,19,21,28,36,46,47,54,57,59,60] which describes beliefs regarding the positive results of HPD use. The main attributes usually characterizing these two cognitive-perceptual factors are listed in Table 1. They have been gathered from an exhaustive literature review carried out on references [6,7,8,9,10,11,12,13,14,15,16,17,18,19,20,21,22,23,24,25,26,27,28,29,30,31,32,33,34,35,36,37,38,39,40,41,42,43,44,45,46,47,48,49,50,51,52,53,54,55,56,57,58,59,60].

Engaging in a health behavior, such as the consistent use of HPDs, is a complicated process which can take time and efforts. However, the temporal dimension is not accounted for in behavioral models classically used to predict this HPD-related health behavior (e.g., HBM, HPM). These behavioral models assume a linear relationship between predictors and behaviors [31] and do not capture the scope of change, which effectively occurs both for comfort judgments and health behaviors. In order to account for behavior change with time, health behavior stage models were thus developed [31] (p. 40). These models conceptualize the temporal dimension of behavioral change through different stages. Behaviors and their predictors change across stages, suggesting that interventions will be more effective when they are tailored to an individual’s current stage of change. The Transtheoretical Model (TTM) [62] is the best-known health behavior change model. It aims to describe how individuals progress toward adopting and maintaining a given behavior. The theoretical framework of this model is based on multiple constructs [62], the following two being of particular interest in this work: the “stage construct” representing a temporal dimension and the “decisional balance” reflecting the individual’s relative weighting of the pros and cons of changing. The concepts of pros and cons are similar to those of benefits and barriers classically used in social cognition models. Therefore, the rare studies applying TTM to predict consistent use of HPDs also mixed it with HPM [6,14,24]. These studies found that barriers to HPD use (with respect to benefits of HPD use) decrease (with respect to increase) across stages: the benefits over barriers ratio increases with time. However, they also show that most workers do not wear HPDs consistently. Extensive efforts remain to be made in order to better understand these benefits and barriers and their origins and find ways to increase the benefits over barriers ratio to such a level that people consistently use their HPDs and are confident that they will not relapse and do not feel the need to behave unhealthily again.

### 1.2. Limitations of Existing Behavioral Models

Despite their utility and interest, existing behavioral models do not explain all HPD-related health behaviors leading to a healthy condition (here, preventing NIHL or, more generally, reducing all the health effects of noise). Indeed, the effectiveness of the HPD does not only depend on its consistent use but also on its correct use, the latter being characterized by the quality of the acoustic seal (or fit) ensuring the necessary protection from external noises [1,63,64]. In other words, an improperly worn HPD is almost useless, and, worse, may cause NIHL in workers who believe they are rightly protected. This false impression can lead them to expose themselves to a higher noise level and for a longer period of time. Even so, consistency of use is the only outcome of existing health behavior models, although the correct use of HPD should be included as an outcome of future models. 

Furthermore, by definition, behavioral models aim to predict human behavior. They thus focus on the characteristics of the individual and on the psychosocial factors of his/her work environment. However, since the health behaviors of interest in this work involve an interaction with a protective device, it is also important to consider the characteristics of the latter, since they can significantly influence barriers and benefits perception (see Table 1) through comfort aspects and thus influence HPD use. In fact, the most common barriers influencing HPD use are also found to be the main discomfort features associated with HPD use [61]. They can then be influenced by characteristics of the environment, the worker (user) and the HPD itself, all together constituting the triad “environment/person/HPD” [65].

Table 2 presents known incompatibilities between HPD characteristics and those of the two other triad components and which were mainly gathered from a reference book [1] (see chapter 11) but also from the grey literature [66], research papers focusing on specific issues associated with HPD use [67,68,69] and a general survey on HPD use [70]. These incompatibilities can generate discomforts and thus lead to preferences for a given type of HPD, or in the worst case to the non-use of the HPD. Most of the perceived barriers and benefits thus highly depend on the HPD type [1] (e.g., earplugs or earmuffs) or model (which may differ, for example, in terms of material or shape). However, the existing behavioral models consider the HPD globally (i.e., without differentiating earplugs from earmuffs and thus without taking into account the HPD characteristics) and only as an output associated with the health behavior (i.e., consistent use of the HPD). This gives the misleading impression that the HPD itself, and consequently the perceived comfort, does not influence the worker’s behavior. Finally, as mentioned previously, behavioral models aim to develop effective interventions to promote the use of HPDs, which constitutes a very important strategy to increase the benefits over barriers ratio. However, another challenging strategy that should not be discarded is the design of more comfortable HPDs, i.e., adapted to the person and his/her work environment, as well as to the interaction between them that will result in the work activity.

### 1.3. Complexity of Comfort-Related Aspects and How They Are Addressed

A deep understanding of HPD comfort and how it affects its use is necessary in order to design and propose more comfortable HPDs. These HPDs should be adapted to the user and his/her work environment. However, comfort is complex and multidimensional, and difficult to define and characterize with both subjective and objective measures, as it undoubtedly involves feelings and emotions that are subjective in nature [72]. 

Three important aspects arise from these observations. Firstly, a wide variety of definitions and constructs of HPD comfort can be found in the literature. This hinders the combination and analysis of data from different studies and thus the proposal of general outcomes on HPD comfort for research, engineering and occupational health practice [61]. Secondly, comfort can be influenced by multiple characteristics of the triad “environment/person/HPD” and be defined by complex interactions between its components [65]. However, an exhaustive list and understanding of all factors affecting (dis)comfort, their relative contribution and influence on HPD use is still missing. Thirdly, the strength of some comfort-related barriers may decrease over a given period of time, referred to as mid-term habituation [65]. This acclimatization time has been observed for both physical and acoustical discomfort attributes and would range from days to months [65]. In any case, the user needs to persist in the behavior to experience such a relief. Unfortunately, this acclimatization time has not been investigated thoroughly and may be difficult to capture and analyze since it may vary from a discomfort attribute to another (e.g., physical pain inside the earcanal or speech intelligibility) and depends on multiple physical and psychosocial characteristics of the triad components and their complex interactions which are still unknown, as mentioned previously. 

Due to all the aforementioned complexities, HPD comfort is usually addressed using different and limited constructs, but also using empirical approaches based on trial and error, both during the HPD design, selection and use phases. Regarding the design phase, only few details can be found in the grey literature. For example, a patent on polymeric roll-down foam earplugs [73] provides quantitative values of some HPD properties (e.g., geometry, recovery time and static mechanical pressure exerted on earcanal walls for roll-down foam earplugs) associated with increased comfort. Manufacturers, however, are not asked to provide the characteristics of the work environment (e.g., temperature, humidity) or of the most compatible user (e.g., earcanal morphology) for the HPD who might favor a comfortable fit with time. As mandated by the U.S. regulation [74], only the noise reduction provided by the HPD (quantified by the Noise reduction rating NRR) is measured and labeled on the product’ packaging. Accordingly, the selection phase at the workplace is mainly based on the worker’ noise exposure level and the NRR of the HPD. As far as comfort is concerned, the Canadian Standards [63,64] advise that employers offer a selection of types of HPD to the workers so that they find out the one that is the most adapted to them and their work environment and can be worn comfortably throughout their shifts. 

As a first step towards improving the understanding of HPD comfort, Doutres et al. [61] proposed a multidimensional construct composed of four dimensions: (a) the ‘physical’ dimension, which is related to user perception resulting from biomechanical and thermal interactions; (b) the ‘acoustical’ dimension, which is related to the modification of the perception of both external and internal noises; (c) the ‘functional’ dimension, which corresponds to the practical acceptability of HPDs and refers to usability, efficiency and usefulness concepts; and (d) the ‘psychological’ dimension, which refers to the well-being and the satisfaction of the user. HPD comfort is then defined as a global perception characterizing the balance between the four dimensions associated with the relationship between the user and his/her HPD in a given work environment. Over the last decades, researchers have proposed comfort models with clear organization [75,76,77] in order to add comfort-driven design methods in the design process of products (such as in the human-centered design approach). From a theoretical perspective, these comfort models allow clear identification of the different factors influencing comfort and their interaction, and/or to explain the (dis)comfort judgment process. From a measurement perspective, they allow one “*to immediately see the gaps in available measurement techniques*” [78] and thus ensure the robustness of subjective comfort evaluation (e.g., through self-reported questionnaires). From a practical perspective, they allow to one “*inform, direct and improve practice*” [79] and help designers and engineers optimize the product by using comfort-related issues in the early phase in the design process. Because of the complexity of the concept of comfort, its definitions and/or comfort theories are generally proposed in very specific frameworks such as nursing [80], working environments [81,82], sitting [72,75,76,83,84] or clothing [78]. Nevertheless, to the authors’ knowledge, there is no comfort model for HPDs.

### 1.4. Synthesis of the Literature

Hearing protectors are commonly offered to workers as last line of defense for preventing NHIL. However, the effectiveness of HPDs is questionable since they are sometimes not used at all or used inconsistently and/or incorrectly. Numerous behavioral models have been proposed in the past to investigate the main factors explaining inconsistent use of HPDs and to develop effective interventions to promote HPD use. These models have shown that (i) “perceived barriers” and “perceived benefits” are strong predictors of the HPD-related health behavior (i.e., consistent HPD use) and (ii) the benefits over barriers ratio increases with time and is associated with a change of behavior towards engaging in the health behavior without relapse. In the authors’ opinion, the existing behavioral models of HPD use nevertheless suffer from two main drawbacks. First, the effectiveness of the HPD does not only depend on its consistent use but also on its correct use. The latter health-behavior should thus also be an outcome of behavioral models. Second, these models focus on the characteristics of the individual and on the psychosocial factors of his/her work environment. The HPD is considered globally (i.e., without differentiating earplugs from earmuffs) and only as an output associated to the health behavior, thus giving the misleading impression that the HPD itself does not influence the worker’s behavior. However, various discomfort aspects associated with HPD use are well known to discourage workers from engaging in a consistent and correct use of HPDs [85]. These discomfort aspects, being also the most common barriers influencing HPD use, are known to depend on physical and psychosocial characteristics of the triad “environment/person/HPD” and to vary over time, delimiting a period known as acclimatization time [65]. Even if a definition of comfort for HPD can now be found [61], the process of (dis)comfort and its evolution with time is still unknown and needs to be investigated. To partly fill this gap, comfort models have been proposed in the past for different products (e.g., chairs, cloths) but there is no comfort model for HPDs. Such a model could be used to better understand the origins of (dis)comforts which are also the perceived benefits and barriers explaining the unhealthy behaviors. As mentioned previously, understanding the unhealthy HPD-related behaviors and their origins is of utmost importance to be able to propose solutions to improve the efficiency of this NIHL protection measure. 

### 1.5. Purpose of the Article: Research Focus

In this paper, the authors propose a holistic model of HPD use in order to provide a framework to better understand all HPD-related health behaviors (i.e., consistent and correct use), their origins and evolutions with time, and ultimately to improve the efficiency of this solution to prevent NIHL. To do so, a comfort model dedicated to HPDs is first developed in order to account for the complex mechanisms from which comfort-related barriers and benefits originate (see Section 2). The comfort model is then integrated in an existing social cognition model already developed for HPDs [29] in order to predict HPD-related health behaviors accounting for the (dis)comfort aspects and the influence of physical and psychosocial characteristics related to the environment, the person and the HPD (see Section 3). Finally, the temporal dimension of behavioral change is taken into account by adding the “stage construct” of the TTM [62] to the previous “time-independent” behavioral model to give the proposed holistic model of HPD use (see Section 4). 

## 2. HPD Comfort Model

This section presents the first step of the construction of the proposed holistic model. The proposed HPD comfort model is shown in Figure 1. It is derived from the previous models developed for clothing by Branson and Sweeney [78] and for general products by Vink and Hallebek [75]. The proposed comfort model starts with the context in which a person uses a HPD in a given work environment (see orange boxes in Figure 1) and consists of four phases (see blue boxes) in the process before “Comfort” (C), “Discomfort” (D) or “feel Neutral” (N) is experienced (see dark grey boxes). 

### 2.1. Comfort Model Phases

The four phases of the proposed comfort model are now presented, and their characteristics are listed in Table 3. The first phase of the Vink and Hallebek [75] model is the “Interaction” phase. It is divided here in two phases involving two different types of interaction between the user and the HPD: the “Fitting/positioning” phase (F) and the Interaction phase (I). This distinction originates from the fact that the product of interest in this work is a protective equipment requiring specific attention prior to its final interaction with the ear it aims to protect. The “Fitting/positioning” phase (F) is the first phase and is part of the usage of the product. It includes all the gestures and care involving the HPD and the body as recommended by the HPD manufacturer in order to maximize the quality of the fit and consequently the protection efficiency. For example, in the case of a “roll-down foam” earplug, the fitting procedure involves compressing (rolling) the earplug between the fingers, pulling the pinna prior to insertion, setting the desired insertion depth and holding the earplug in place with a fingertip for a few seconds to allow the earplug to expand. The Interaction phase (I) then occurs when the HPD is fitted and in place. During this phase, the final contact between the HPD and the outer ear (e.g., inside the earcanal for earplugs or around the ear for earmuffs) is established. The HPD applies a mechanical pressure at the contact surface with the human tissues. An interaction also occurs in the acoustical dimension as the acoustic pressure at the eardrum, induced either by external or internal sources, is modified due to the presence of the HPD. These interactions result in “internal Human body effects” (H) such as tissue deformation and the compression of nerves and blood vessels in the physical dimension. In the case of the acoustical dimension, the acoustic energy reaching the inner ear through the outer, middle and/or inner ear pathways is converted to electrical impulses sent to the brain. Physical and acoustical stimuli also influence physiological responses and result, for example, as a modification of the heart rate or blood pressure [86]. These effects are then perceived in the fourth phase (P). The HPD can be perceived as a foreign object which modifies the perception of internal sounds (e.g., user’s own voice) and the external noise environment (e.g., useful machinery signals).

### 2.2. Comfort Model Outcomes

Finally, the perceptions (of phase (P)) are judged as comfortable (C), uncomfortable (D) or neither (N). According to the construct of HPD comfort proposed by Doutres et al. [61], the (dis)comfort judgments then belong to the following four dimensions: physical, functional, acoustical and psychological. The comfort and discomfort judgments are listed in Table 3. These lists are not exhaustive. They consist of (i) the main earplug comfort attributes identified in reference [61], (ii) common discomforts mentioned in reference books [1] or standards [63,64] and (iii) barriers and benefits identified in behavioral studies and related to comfort judgments as shown in Table 1.

### 2.3. Comfort Model Inputs

As proposed by Branson and Sweeney [78], the inputs to the comfort model consist of intrinsic characteristics of the triad “environment/person/product” and are categorized in physical or psychosocial dimensions. In this work, the triad has been adapted to the context of HPDs [65] and all the physical (listed in Table 4) and psychosocial (listed in Table 5) characteristics have been gathered from a literature review including both comfort [61,65] and behavioral studies on HPDs [6,7,8,9,10,11,12,13,14,15,16,17,18,19,20,21,22,23,24,25,26,27,28,29,30,31,32,33,34,35,36,37,38,39,40,41,42,43,44,45,46,47,48,49,50,51,52,53,54,55,56,57,58,59,60].

The inputs of the proposed comfort model also cover the main inputs of the existing comfort models (i.e., “Person”, “Product”, “Task and usage” and “Gratification level and emotions”) [75,87,88] but are organized differently. Characteristics of the “Person” and “Product” of these models are here identified as main components of the triad. However, “Task and usage” [75,87,88] and “Gratification level and emotions” [87,88] are included here as aspects of the “Environment” component of the triad, and more particularly in its psychosocial dimension (see Table 5). According to Naddeo [88], “Gratification level and emotions” refer to “*the set of work characteristics and the emotional state that contribute to the satisfaction/dissatisfaction of the worker (job position in organization chart, working shifts, gratification, salary and so on) and is widely related to the general environment*”. The corresponding characteristics are integrated here as situational influences (see Table 5). “Task and usage” refers to “*all the task or the use that human can do*” during the interaction with the product [88] and includes time aspects (duration, frequency, time of the day) [76]. As mentioned previously, “Task and usage” are also considered within the framework of a psychosocial approach (i.e., included in the psychosocial dimension presented in Table 5), which is relevant in authors’ opinion, since this approach takes into account the person performing his/her required work task and his/her social interactions in the work environment. 

By using the concept of triad categorized in these two main dimensions, the authors also concur with the associated analyses of Branson and Sweeney [78] which are briefly recalled later. First, this organizational structure is very important since it allows for quick identification of gaps in measurement techniques. Second, it is believed that connections or interactions can occur between the lists, either between characteristics of the two dimensions or between characteristics of the triad components. This is represented by using dashed black lines within the model in Figure 1. Third, the lists of characteristics provided in Table 4 and Table 5 are not exhaustive. They are given as a basis for discussion and are expected to be completed over time.

### 2.4. Links between Comfort Model Components

The various links between the model components are now briefly presented (see black arrows in Figure 1). The Fitting/positioning phase (F) is influenced by characteristics of both physical and psychosocial dimensions of the triad. Physical characteristics of the person and of the HPD will influence HPD fitting. For example, push-to-fit foam or premolded earplugs may not be adapted to everyone’s earcanal and can lead to important air leaks during the Interaction phase (I) for a user having large earcanals [1]. The Fitting phase (F) can also be influenced by the user’ past experience and habits, the type of training, the frequency of training, his/her degree of acculturation [20], but also by the worker’ activity: a worker who needs to insert/remove a HPD regularly during work or to wear other protective equipment may not be able to fit the HPD correctly. 

The perceptions (P) originating from the internal Human body effects (H) may also be directly influenced by multiple psychosocial characteristics of the triad and related to the individual’s (or worker’s or user’s or subject’s) expectations regarding HPDs (e.g., prior experience with HPD, expectation of perceivable comfort), his/her lifestyle, social environment at work or cultural background. Some psychosocial characteristics related to “task and usage”, such as work duration, the type of work or the necessity to hear useful signals (machines/engines, alarms) can also affect comfort judgment. The triad characteristics belonging to the psychosocial dimension and affecting perceptions (either at the conscious or the unconscious level) thus play the role of the “stored modifiers” in the comfort model of Pontrelli [89] or “filter” in the model of Branson and Sweeney [78]. Accounting for their influence illustrates the complexity of the comfort judgement. Indeed, a given HPD worn in a given work environment can result in different comfort judgments due to the uniqueness of the user. 

The proposed HPD comfort model also includes a feedback loop if discomfort (D) is too important. It is connected to the Fitting/positioning phase (F) which could be modified intentionally by the user. For example, earplug’ users feeling pain in the earcanal or difficulty in hearing useful sounds commonly pull the HPD out of the earcanal to reduce the mechanical pressure and decrease the acoustic seal performance respectively. The feedback loop is also connected to the triad characteristics which allow adjustments by the user. For example, the user can damage/alter the physical properties of the HPD [1], adapt some postures and/or movements during his/her work activities or decrease the communication duration with colleagues in order to improve HPD comfort (or to reduce discomfort).

### 2.5. Comfort Model Evaluation Instruments

As suggested by Naddeo et al. [77], the process leading to the comfort judgement can be measured at multiple phases. It is here adapted to the proposed HPD comfort model of Figure 1 (see dash-dotted black arrows). First, the characteristics (psychosocial and physical) of the components of the triad can be measured subjectively using questionnaires and observations and/or objectively using sensors. In order to improve our understanding of perceived HPD (dis)comfort and to reduce comfort measurement variability, it is advised to quantify as many triad characteristics as possible and use them as independent or control variables [65]. At the Fitting/positioning phase (F), questionnaires or observations are commonly used. At the Interaction phase (I), the use of field attenuation estimation systems [90] (FAES), microphones or medical images (e.g., from magnetic resonance imaging MRI scanners [91]) is possible. Visual inspections could also be used for assessing the fit quality, but this is a much less robust method [65]. At phase (H), various sensors may be used to assess physiological and/or neurophysiological responses of the body. At the Perception phase (P), questionnaire or objective tests (such as hearing in noise test) may be performed. Finally, the use of questionnaires enables assessment of a person’ experience of Comfort (C), Discomfort (D) or Neutral feeling (N). 

## 3. Time-Independent Model of HPD Use

This section presents the second step of the development of the proposed holistic model, based on the integration of the HPD comfort model (see Figure 1) into the behavioral model of Hong et al. [29]. The time-independent model of HPD use shown in Figure 2 is thus obtained. 

### 3.1. HPD Use Model Components

The Hong et al. [29] model was selected as a basis of the proposed holistic model because it is based on social cognition theory and models (e.g., HPM, HBM) which are the most widely used and proven to predict HPD-related health behavior (i.e., consistent use of HPD, see Section 1.1). This model uses cognitive-perceptual factors as proximal determinants of behavior. Other “modifying factors” may also influence behavior, directly and/or indirectly, through proximal cognitive-perceptual factors. These distal (modifying) factors usually include demographic (e.g., age, gender, ethnicity) and psychosocial characteristics (e.g., interpersonal support, organizational health support, perceived HPD availability).

In addition to the “perceived benefits” and “perceived barriers” cognitive-perceptual factors discussed in the introduction of this paper, this model accounts for the three following factors also pointed out in the literature as important predictors influencing HPD use [7,8,9,12,19,21,29,36,40,50,54,92,93]: “perceived self-efficacy” which characterizes the worker’ confidence in their ability to use HPDs correctly, “perceived susceptibility to hearing loss” and “perceived severity of hearing loss”. The arrangement of the cognitive-perceptual factors has been slightly modified compared to the original model [29] in order to account for some recent modifications of the HPM model [31]. Here, the perceived benefits and barriers can now be influenced by the three other cognitive perceptual factors (see Figure 2). It is worth noting that while the Hong et al. [29] model has been chosen here, any individual or community model to promote health could have been used as long as perceived barriers and benefits of HPD use are included as factors explaining the health behaviors.

The integration of the HPD comfort model (see Figure 1) is performed at its two extremities. First, the modifying factors of the original behavioral model of Hong et al. [29] are rearranged according to the organizational structure of Branson and Sweeney [78] (i.e., characteristics categorized in triad components having two dimensions) (see Section 2.3 and Section 3.2). Second, its outputs (i.e., “Comfort” (C), “Discomfort” (D) and “feel Neutral” (N)) are merged with the “perceived benefits” and “perceived barriers” cognitive-perceptual factors because of the ties between the two concepts as illustrated in Table 1. The neutral output of the comfort model (N) is set as an input for both “perceived benefits” and “perceived barriers”. Benefits and barriers can then be based on personal outcomes from direct personal experience with the behavior or vicarious experience through observing others engaging in the behavior [31] (p. 37). Nevertheless, only benefits and barriers experienced by the user outcome from the comfort model since they originate from physical and psychosocial interactions between all “environment/person/HPD” triad components. Anticipated benefits and barriers based on the observations of others, and which do not involve any physical interactions between the user and the HPD are accounted for using a direct link between the modifying factors and the cognitive-perceptual factors (i.e., bypassing the HPD comfort model). 

### 3.2. HPD Use Model Inputs

The inputs of the comfort model (see Table 4 and Table 5) are used as inputs for the time-independent model (and later for the holistic model described in Section 4). Consequently, some psychosocial factors expected to influence the HPD-related behavior are added compared to the original model of Hong et al. [29]. Some of them, such as “Prior related behavior” (see Table 5) concur with the recent revision of the HPM [31] which highlights this factor by using it as a distinct input in the model. According to Pender [31] (p. 36): “*Research indicates that often the best predictor of behavior is the frequency of the same or a similar behavior in the past*”. The great importance of the previous HPD related behaviors will be emphasized in Section 4 dedicated to the holistic model which accounts for the scope of change. 

### 3.3. HPD Use Model Outcomes

Compared to the existing behavioral model for HPD use, the outcome of the present model is not restricted to the consistent use of HPD but also includes its correct use (see pale green box in Figure 2). Health behaviors imply that the HPD is always fitted properly and worn 100% of the time when exposed to hazardous noises. Short-term benefits on health are also expected from these health behaviors (e.g., reduced headaches, fatigue and stress induced by the noise exposure) although they are not directly targeted by the existing workplace noise regulations. 

### 3.4. Links between HPD Use Model Components

The links between the model’s components are presented with black arrows in Figure 2. According to the proposed HPD comfort model, the modifying factors influence the Fitting/positioning (F) and Perception (P) phases. The modifying factors are themselves influenced by the discomfort judgments through the feedback loop. According to the behavioral model, the modifying factors also impact HPD-related health behaviors either directly or indirectly through the proximal cognitive-perceptual factors. Finally, in agreement with the revised HPM model, a link is added to account for the possible influence of three cognitive-perceptual factors (i.e., “perceived self-efficacy”, “perceived susceptibility to hearing loss”, “perceived severity of hearing loss”) on the Perception phase of the HPD comfort model. For example, a person perfectly aware of his/her susceptibility to NIHL could unconsciously judge in a more favorable manner a physical discomfort originating from the interaction of the HPD and his/her outer ear.

### 3.5. HPD Use Model Evaluation Instruments

Multiple evaluation methods and indicators for assessing the HPD consistent use have already been proposed and analyzed (e.g., supervisor report, self-report, direct observation) [7,94,95,96,97]. The assessment of correct use (i.e., fit quality), on the contrary, would be new in the operationalization process of behavioral model of HPD use. It could be determined (i) from an external person based on visual inspections, or preferably using (ii) a fit-testing systems technically referred to as field attenuation estimation systems (FAES) [65] or (iii) an instrumented HPD providing its real-time attenuation during the entire work shift.

## 4. Holistic Model of HPD Use

In order to take into account for the effect of time on HPD comfort and HPD-related health behaviors, and considering the ties between the three concepts “pros, benefits and comforts” and their antagonists “cons, barriers and discomforts”, the authors propose the holistic model shown in Figure 3. This model is based on adding the “stage construct” of the TTM (see Section 1.1) to the time-independent model described in Section 3.

### 4.1. Holistic Model Stages

According to the “stage construct” of the TTM, the health-related behavior change progresses through the five following stages [62]: precontemplation, contemplation, preparation, action and maintenance. The general definition of these stages is briefly presented in Table 6. A sixth stage, called termination, is also part of the TTM and refers to a stage in which people are confident that they will not relapse and do not feel the need to behave unhealthily again. This stage is often excluded from health promotion programs because it is rarely achieved, and people stay in the maintenance one. Accordingly, it will also be omitted in the proposed holistic model. 

When applied to HPDs, staging is carried out from the following measures [6,14,24]: (i) the actual use of HPDs (i.e., the percentage of time the HPD is used when exposed to high noise levels), (ii) the commitment of using HPDs in the future and (iii) the duration of previous related behavior. Table 6 also presents the stages of hearing-protection behavioral change as proposed by Kalampakorn [6] (see Appendix I in ref. [6]) and effectively used by Hong et al. [14]. The five stages of hearing-protection behavioral change differentiate individuals who do not use HPDs (precontemplation and contemplation stages) with those who use them inconsistently (preparation stage) or consistently for a given period of time (action and maintenance stages). It is worth noting that the staging algorithm varies between studies. This is most probably because it has been adapted to the actual use of HPD for a specific population. For example, the threshold for consistent use can vary across studies. It was set to 90% for construction workers in [14] and to 100% for factory workers in [24]. In the same way, the period of time required to achieve maintenance varies from study to study. Citing the same examples, it was taken equal to six months in [14] and one month in [24]. Anyhow, the stages make it possible to distinguish the situation of the workers according to three temporalities of experience with HPDs: their past, current and future experience. 

### 4.2. Integration of the Time-Independent Model

In agreement with the previous studies applying the TTM to the HPD-related behaviors [6,14,24], the time-independent behavioral model of Section 3 is applied to all aforementioned stages (see Figure 3). However, only the two constructs of interest in this work (i.e., perceived benefits and barriers) are presented in each stage for the sake of conciseness. The comfort model is integrated only in the preparation, action and maintenance stages since they involve wearing the HPD. In agreement with Prochaska et al. findings on various health behaviors [62], also confirmed in the case of the HPD-related behavior [6,14,24], the two cognitive-perceptual factors of interest (i.e., barriers, and benefits) differ significantly between stages. The size of the boxes used in Figure 3 visually represents their evolution across the stages: while the benefits (pros) increase, the barriers (cons) decrease.

### 4.3. Holistic Model Inputs and Outcomes

All stages are influenced by the triad characteristics (modifying factors) already presented in Section 2.3 and Section 3.2. From the preparation stage, the outcome of each stage corresponds to the behavior listed in Table 6, but also includes the correctness and the consistency of use according to the holistic model outcomes. Each stage outcome can then act as a “prior related behavior with HPD”, previously considered as a psychosocial characteristic of the person (see Table 5), and which may influence the next phase in case of progress (see green arrows) or the previous phase in case of relapse (see red arrows). This is in agreement with the revised HPM [31] which stresses the importance of prior related behavior on the health behavior (either directly or indirectly through perceptions of self-efficacy, benefits, barriers, and activity related affect).

## 5. Conclusions and Discussion

### 5.1. Summary

In this work, a holistic model of HPD use was proposed in order to provide a framework to better understand HPD-related health behaviors, their origins and evolution over time and, ultimately, to improve the efficiency of this solution to prevent NIHL. To do so, a comfort model dedicated to HPDs was first developed in order to account for the complex mechanisms from which comfort-related barriers and benefits originate and which have already been identified as major predictors of HPD-related health behaviors. These complex mechanisms successively involve (i) the fitting/positioning of the HPD, (ii) the interactions between the HPD and the outer ear, (iii) the internal body effects resulting from these interactions (e.g., tissue deformation, compression of nerves), (iv) the perception of the latter effects and (v) the judgments of (dis)comfort. The comfort model was then integrated into an existing social cognition model already developed for HPDs but in which modifying factors inputs have been categorized according to the triad components “environment/person/HPD” and their psychosocial and physical dimensions. In order to account for the fact that changing the behavior towards a consistent and correct use of HPDs is a complex and long process, the (time-independent) behavioral model (see Figure 2) was finally integrated within the “stage construct” of the transtheoretical model to give the proposed holistic model of HPD use (see Figure 3). The proposed HPD comfort model is then involved in the last stages of change where the benefits over barriers ratio should increase significantly enough to enable the engaging and the maintaining of health behaviors.

### 5.2. Comparison with Other Research

The scope of this holistic model is broader than existing behavioral models of HPD use. Indeed, the outcome of the proposed holistic model is not restricted to the consistent use of HPD but also includes its correct use which is equally important for effective hearing protection. Furthermore, unlike previous behavioral models, the proposed holistic model accounts for the possible influence of both physical and psychosocial characteristics of all the triad components, and thus for the fact that the HPD itself can influence the worker’s behavior through (dis)comforts aspects. Undoubtedly, physical characteristics of the triad components affect (dis)comfort through a direct input of the Fitting/positioning phase (F) of the HPD comfort model (see Figure 1). Thus, they indirectly affect HPD use through comfort and discomfort judgments (e.g., pain due to physical contact between the user and the HPD) and which are commonly referred to as benefits of use and barriers to use, respectively (see Table 1). It is thought that the inclusion of physical characteristics of the triad components in behavioral models could improve the development of future interventions giving more weight to the incompatibilities (to be avoided) between them. Finally, the proposed holistic model accounts for the temporal dimension, thus allowing to capture of the scope of change, which according to the literature, effectively occurs both for (dis)comfort judgments and HPD-related health behaviors.

Concerning more specifically the HPD comfort model, as mentioned previously, the proposed model of Figure 1 is heavily inspired by existing comfort models [75,78,87]. However, it has been adapted to the context of protection equipment requiring specific attention prior to its final interaction with the person’s ear. To do so, the model starts with a “Fitting/positioning” phase (F) to account for all the gestures and care involving the HPD and the body as recommended by the HPD manufacturer in order to maximize the quality of the fit. The inputs of the comfort model consist of intrinsic characteristics of the triad “environment/person/product” and are categorized in physical or psychosocial dimensions. It is worth noting that the large number of psychosocial characteristics considered as inputs of the comfort model (see Table 5) mostly come from the considerable research on worker behaviors related to HPD use and briefly summarized in the introduction to this paper (see Section 1.1). This is an advantage over existing comfort models proposed for other products which usually put the focus on physical characteristics of the triad and do not provide many details on the influence of psychosocial aspect.

### 5.3. Strengths and Limitations

In the authors’ opinion, the proposed holistic modeling approach aims to clarify the key concepts (i.e., the triad and its physical and psychosocial dimensions, (dis)comfort and its four dimensions, and the ties with barriers and benefits). It also provides a framework to allow visualization of the whole, i.e., from a detailed description of the triad components and their complex interactions up to (dis)comforts and to the HPD-related health behaviors and their evolution with time. According to the various studies on TTM applied to HPD use [6,14,24], surpassing the preparation stage for achieving maintenance is challenging and requires significant increase in the benefits over barriers ratio. This could be achieved using two main strategies: (i) designing HPDs which are comfortable and/or provide a quick acclimatization period for the user in his/her given work environment (i.e., lower barriers), (ii) performing interventions (e.g., training, HPD selection) tailored to an individual’s current stage of change (i.e., higher benefits). Both strategies require a better understanding of the temporal evolution of benefits/comforts and barriers/discomforts for each dimension [61] and of their influential factors (physical or psychosocial). Furthermore, as stated previously, the design of comfortable HPDs requires comfort-driven design methods based on comfort models capable of objectifying comfort judgments. It is thought that the holistic model provides a dedicated framework to reach these challenging goals.

By merging comfort and behavioral models, the proposed holistic model allows a comprehensive view of all potential factors affecting comfort and health behaviors. However, the associated drawback is the increase of the experimental cost of the field studies. Indeed, in order to increase our understanding of comfort judgments and HDP-related health behaviors using the proposed framework, it is advised to quantify as many triad characteristics as possible [65] (see Table 4 and Table 5 and Section 2.5 and Section 3.5). Measurements of the holistic model’s inputs and outcomes should also be carried out at different times to capture and better understand the scope of change, including the acclimatization time for discomfort issues. Furthermore, adding an outcome such as the HPD fit quality to behavioral models of HPD use obviously complexifies the operationalization of the proposed model. The assessment of the HPD consistent use is already complex but has been investigated for many years (see Section 3.5). Adding the assessment of HPD correct use will obviously increase the experimental cost of research, and the most relevant evaluation instruments for field studies (in terms of applicability and cost) remain to be found.

### 5.4. Future Research

The proposed holistic model of HPD use could be used in the short term to identify the most important characteristics of the “environment/person/HPD” triad components and how they affect (dis)comfort and HPD-related health behaviors. It could then be used by psychologists, audiologists, ergonomists, and/or occupational hygienists to develop and plan interventions in order to increase the benefits to barriers ratio related to the HPD use while giving more importance to the interactions between triad components (thus including the characteristics of the HPD) and to the effect of time. For their part, engineers, ergonomists, and manufacturers could use the proposed holistic and comfort models to objectify the main discomfort attributes via dedicated models and use them in the early phases of the design process to develop comfortable HPDs adapted to the characteristics and requirements of groups of individuals.

## Figures and Tables

**Figure 1 ijerph-19-05578-f001:**
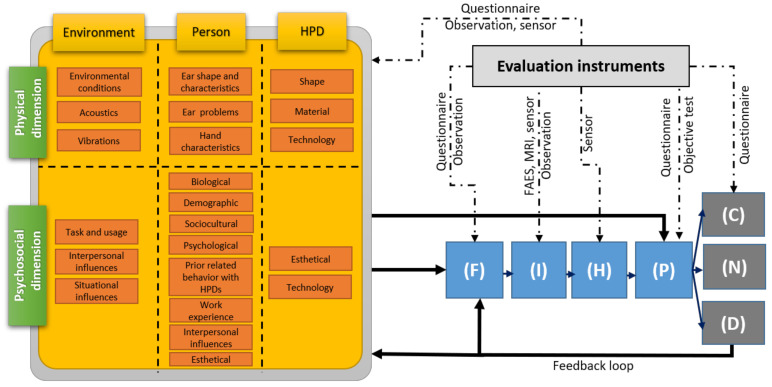
HPD comfort model. Acronyms used in this figure: “Fitting/Positioning” phase (**F**), the “Interaction” phase (**I**), the “internal Human body effects” phase (**H**) and the “Perception” phase (**P**); Comfort (**C**), Discomfort (**D**) or feel Neutral (**N**); Hearing Protection Device (HPD), Field Attenuation Estimation System (FAES), Magnetic Resonance Imaging (MRI).

**Figure 2 ijerph-19-05578-f002:**
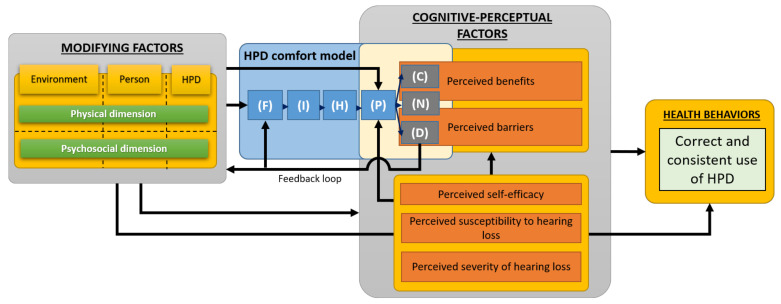
Time-independent model of HPD use.

**Figure 3 ijerph-19-05578-f003:**
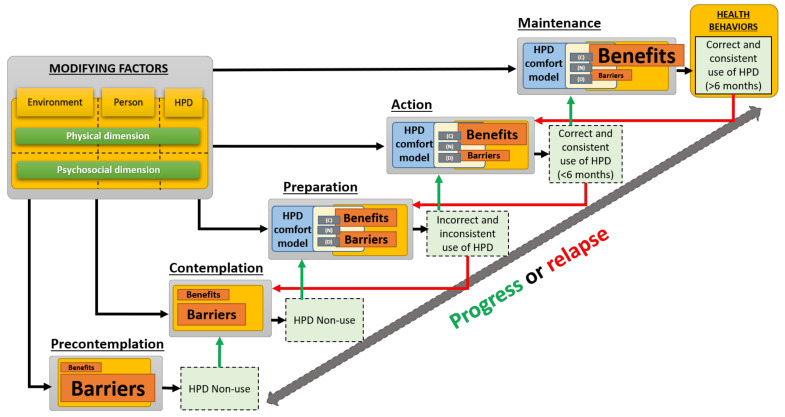
Time-dependent holistic model of HPD use.

**Table 1 ijerph-19-05578-t001:** Attributes of perceived barriers and benefits of HPD use according to behavioral studies [6,7,8,9,10,11,12,13,14,15,16,17,18,19,20,21,22,23,24,25,26,27,28,29,30,31,32,33,34,35,36,37,38,39,40,41,42,43,44,45,46,47,48,49,50,51,52,53,54,55,56,57,58,59,60] and their correspondence to Doutres et al.’s [61] comfort dimensions.

	Attribute	Comfort Dimension ^1^
**Barriers**	Aspects related to the mechanical contact between the HPD and the body (e.g., unpleasant mechanical pressure and irritation)	Physical
	Communication difficulties (hearing and speech)	Acoustical
	Difficulty hearing useful machine sounds and danger signals	Acoustical
	Difficulty in inserting the HPD	Functional
**Benefits**	Protect hearing/prevent hearing loss	Functional
	Prevent tinnitus (ringing in the ears)	Physical/ Functional
	Reduce headaches and fatigue	Physical
	Avoid noise annoyance	Functional/Acoustical/psychological
	Enhance communication	Acoustical
	Enhance ability to hear useful machinery noise	Acoustical

^1^ Attributes may belong to multiple comfort dimensions since the boundaries between the dimensions are not hermetically sealed.

**Table 2 ijerph-19-05578-t002:** Known incompatibilities between HPD characteristics and those of the two other triad components.

**HPD/Person**	Every disposable earplug cannot fit correctly in every earcanal, and every earmuff is not suited for all head shapes [1,66]
	Earplug use is not recommended in the presence of an earcanal medical condition (e.g., infections, eczema) [1,67]
	Earmuff protection efficiency can be affected by the presence of hair, eyeglass temples or caps [1]
	Earplugs are preferred by individuals favoring discrete hearing protectors and/or concerned with their hairstyle [1]
**HPD/Work environment**	Earmuffs can be difficult to wear in hot environments [1,67,68]
	Over-attenuating HPDs can block important environmental sounds such as alarm signals, machine sound and colleagues’ speech [1]
	Earmuffs reduce the localization of the noise sources more than earplugs [69]
	Earplug insertion and removal can be time consuming and incompatible with work tasks when many removals are required during the work shift (and thus, earmuff are preferred in this case [1,70])
	Earmuffs can interfere with movements when work is carried out in a tight space [1]
	Compared to earmuffs, earplug efficiency is more dependent on user training, skill and motivation [1]
	HPD requiring a periodic re-positioning is not adapted to work tasks based on a fast pace of repetitive movements [71]

**Table 3 ijerph-19-05578-t003:** Phases and outcomes of the proposed HPD comfort model.

Phases				Outcomes	
Fitting/Positioning (F)	Interactions (I)	Internal Human Body Effects (H)	Perceived Effects (P)	Comforts (C)	Discomforts (D)
**Physical/Functional dimension** *Earplugs*: Pull the pinna prior to insertion. Compress the foam earplugs prior to insertion. Choose the insertion depth. Ensure that hair is not inserted inside the earcanal together with the earplug Add lubricant over custom molded earplug. *Earmuffs*: Remove all hair underneath ear cushions. Seal the cushions firmly against the head. Ensure that the earcups are not fitted crooked or askew over the ears.	**Physical dimension** *During the insertion or removal of the HPD*: Large normal and shear stress applied to the ear (pinna, concha and earcanal…), tissue deformation. *Once the HPD is worn:* added static mechanical pressure distributed in the contact area between person and HPD, tissue deformation. **Acoustical dimension** Modification of the acoustic pressure at the eardrum created either by an external acoustic source or internal mechanical source (e.g., voice...).	**Physical dimension** Compression of nerves and blood vessels. **Acoustical dimension** Conversion of the acoustical and vibratory energy transmitted to the cochlea through airborne and structure-borne pathways to electrical impulses in the auditory nerves.	**Physical dimension** Perception of a foreign object in contact with the ear. **Acoustical dimension** Modification of the perceived noise/sound created either by an external acoustic source or internal mechanical source (e.g., voice).	**Functional dimension** Protect hearing/prevent hearing loss, avoid noise annoyance, stay in position. **Acoustical dimension** Reduction of the useless external noise, ease of communication, ability to hear useful machinery noise and danger signals. **Psychological dimension** Trust, habituation, satisfaction.	**Physical dimension** Pain, irritation. **Functional dimension** Difficulty in fitting/positioning, annoyance, unhandy, unstable, intrusion (inhibit head and body movements). **Acoustical dimension** Difficulty in communication, difficulty in hearing useful machinery noise and danger signals, difficulty in localizing sounds, occlusion effect. **Psychological dimension** Esthetical concerns, isolation from the external environment.

**Table 4 ijerph-19-05578-t004:** Physical characteristics of the triad “environment/person/HPD”.

Work Environment	Person	HPD
**Environmental conditions**: air temperature, relative humidity, atmospheric pressure, air quality (dust). **Acoustics**: noise environment (e.g., spectrum, noise level, noise type, direction), presence of useful acoustic signals (e.g., alarms, machine, discussions with colleagues) and their characteristics (e.g., spectrum, noise level, noise type). **Vibrations**: presence of vibrations from hand tool, ground.	**Ear characteristics**: size and shape of the pinna and external auditory canal, flexibility of the pinna and external auditory canal, auditory canal hairiness, propensity to the build up of earwax. **Ear problems**: ear infections, otitis, eczema, hearing loss, tinnitus. **Hand characteristics**: hand dominance, hand dexterity. **Susceptibility to the effects of noise exposure.** **Other attributes that may interact with the use of HPD** (related or not to esthetical aspects): short or long hair, wear eyeglasses, cap, hat.	*Earplugs*: **Design**: shape (e.g., cylinder, bullet shape, conical, custom molded…) size, corded/uncorded, with stem/without stem, need for lubricant. **Material properties**: softness, weight, texture, heat transfer properties, moisture/vapor transfer properties, made of inert non-toxic substance, expansion time after compression, resistance to humidity. **Embedded technology**: presence of acoustic filters, active vs. passive. *Earmuffs*: **Design**: headband, cushion, cup. **Material properties**: cushion and headband stiffness, cup weight, texture, heat transfer properties, moisture/vapor transfer properties, made of inert non-toxic substance. **Embedded technology**: active vs. passive.

**Table 5 ijerph-19-05578-t005:** Psychosocial characteristics of the triad “environment/person/HPD”.

Work Environment	Person	HPD
**Task and usage**: type of work (manual, non-manual, mixed), physical activity (body, head or jaw movements), type of equipment/tools used, necessity to wear other personal protective equipment, necessity to insert/remove regularly HPD during work (task-related or noise exposure-related), necessity to hear useful signals (e.g., machines/engines, alarms), necessity to communicate with colleagues (teamwork), time aspects (e.g., work duration, full time or part time paid employees, noise exposure duration). **Interpersonal influences (peers, co-workers, hierarchy)**: social models, social norms, interpersonal support. **Situational influences**: workplace rules (mandatory hearing protection policy), perceptions of accessibility and availability of HPD, perception of exposition of high noise levels, organizational support for health, HPD training (existence and duration (in hours)), hearing examination/re-examination, job position in organization chart, gratification, salary, hearing protection requirement status, plant site.	**Biological**: gender. **Demographic**: age, ethnicity. **Sociocultural**: acculturation, education, immigration status, socioeconomic status. **Psychological**: personality, self-esteem, self-motivation, perceived health status, perceived hearing status, self-experienced hearing symptoms (tinnitus and noise sensitivity), low frustration tolerance (LFT). **Prior related behavior (previous experience with HPDs)**: experience with HPD use, preferred HPD, ideal HPD, expectations regarding HPD comfort/discomfort, care of HPD (e.g., clean, regular inspection, change damaged, compromised or aged parts). **Work experience**: years trade experience, years at the plant, years of work in a noisy environment, total years worked, whether prior workplace was noisy, unionized or not. **Interpersonal influences (family, friends)**: social models, social norms, interpersonal support. **Esthetical**: concerned with hairstyle.	**Esthetical**: discreetness, color, aesthetic design (e.g., earplug having a screw shape), attractiveness of the product (e.g., custom molded, active products). **Other**: availability and quality of the instructions for proper fit.

**Table 6 ijerph-19-05578-t006:** Stages of change of the Transtheoretical model.

Stage of Change	Brief Definition from [62]	Application to HPD-Related Health Behaviors from [14]
Precontemplation	No intention to take action within the next 6 months	Workers are not using HPDs and have no intention to do so
Contemplation	Intends to take action within the next 6 months	Workers are not using HPDs but have intention to do so
Preparation	Intends to take action within the next 30 days and has taken some behavioral steps in this direction	Workers use HPDs but inconsistently
Action	Changed overt behavior for less than 6 months	Workers use HPDs consistently during a continuous period from 0 to 6 months
Maintenance	Changed overt behavior for more than 6 months	Workers use HPDs consistently for more than 6 months

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
