# Peer review of "Towards a Holistic Model Explaining Hearing Protection Device Use among Workers"

_ijerph, 2022, doi:10.3390/ijerph19095578_

Round 1
Reviewer 1 Report
The paper entitled “Towards a holistic model explaining hearing protection device use among workers” deals with a very interesting topic, and it included interesting ideas.
However, I have the following comments that hopefully help the authors improve their paper:
- I suggest to the authors a section dedicated to literature review where should analyse the existing works in the way to show the gap in the literature compared to this work. It would be better if authors can have a table comparing the closely related works on various dimensions and clearly showing the contribution of the paper.
- What are the limitations of the study in terms of the proposed method, data used, approaches, and/or analysis?
- The authors should convince the readers of this journal, that their contribution is so important. These issues deserve a deeper discussion: What are the managerial implications from this work? What are the implications for theory and practice? How decision or policy makers could benefit from this study.
- It could be interesting to discuss in the conclusion part, the perspectives, and the improvements of this work.
- As usual a final thorough proof-reading is recommended.
I encourage the author to think along those questions and to develop this work further along those lines.
Reviewer 2 Report
The paper proposes a holistic model of hearing protection devices, which is an important topic. The paper is nicely prepared.
I suggest adding research limitations to the conclusion section, along with directions for further research. Furthermore, the authors could point out the strengths of the proposed model and highlight its advantages over existing models.
I hope that my suggestions will help to improve your paper.
Reviewer 3 Report
This work is a meaningful and significant study to occupational health research. The explanation of the holistic model is detailed; however, the discussion section is still weak. Several concerns were mentioned below.
Lines 42, you mentioned that workers in different sectors (i.e., construction, factory, agriculture, entertainment or service…) use HPDs to protect themselves from noise environment. The HPDs employed in different sectors, for example, expandable foam plugs, muffs, inserts, and plugs, are different. Whether different types of HPDs will influence the inconsistent use as well as the factors affecting the inconsistent use. The types of HPDs used in different sectors can be explained and how these factors can be linked to your content can be described in the section 1.1.
In section 1.1, a number of models related to health behaviours had stated, so how to ensure that all these kinds of models were explained in this study and all attributes of perceived barriers and benefits of HPD use? The way to identify the attribute and consolidate the information related to HPD use can be demonstrated and the content of Table 2 as well.
The time-independent model was used to integrate the HPD behavioural use as a holistic model. It is important to point out the underlying reasons of using the time-independent model for the integration. Why this model is better than other relevant models? What is the linkage between time-independent model and HPD use? Any previous related studies also use this model to describe the relevant behaviours. If yes, these studies can be clearly stated and explained.
This study is significant to the field of occupational health. However, the important contributions of the findings of this study were not comprehensively highlighted. Section 5 conclusions and discussions can be separated into two sections. So in the discussion section, the theoretical and practical contributions of this study can be demonstrated, the strength of the suggested model can be explained and the limitation of this study can be described.
Round 2
Reviewer 1 Report
The manuscript has significantly improved as compared to the previous version. Indeed, the authors tried to improve it, and the main weaknesses are solved.
Thus, in my opinion, the manuscript is recommendable for publication.
Reviewer 3 Report
Accept